# Nontraumatic Headache in Adult Emergency Patients: Prevalence, Etiologies, and Radiological Findings

**DOI:** 10.3390/jcm9082621

**Published:** 2020-08-12

**Authors:** Nadja Handschin, Maria Oppliger, Alex Brehm, Marios Psychogios, Leo Bonati, Christian H. Nickel, Roland Bingisser

**Affiliations:** Emergency Department, University Hospital, CH-4031 Basel, Switzerland; nadjamarina.handschin@usb.ch (N.H.); maria.opliger@stud.unibas.ch (M.O.); alex.brehm@usb.ch (A.B.); marios.psychogios@usb.ch (M.P.); leo.bonati@usb.ch (L.B.); christian.nickel@usb.ch (C.H.N.)

**Keywords:** headache, risk factor, emergency medicine, neuroimaging, intracranial abnormality, mortality

## Abstract

The aim of this study was to measure prevalence, to describe underlying etiologies, and to assess radiological findings, focusing on significant intracranial abnormality (sICA). This was a prospective study of unselected adult patients admitted to the emergency department (ED) in a tertiary care hospital where all presenters were systematically interviewed about their symptoms. We attributed nontraumatic headache with neuroimaging to four groups: Normal or no new finding, extracranial abnormality, insignificant intracranial abnormality, or significant intracranial abnormality. sICA was defined as “needing acute therapy”, “needing follow-up neuroimaging”, or “clinically important neurological disorder”. Among 11,269 screened ED presentations, the prevalence of nontraumatic headache was 10.1% (1132 patients). Neuroimaging (cCT and/or cMRI) was performed in 303 patients. Seventy (23.1% of scanned; 6.2% of all headache patients) patients had sICA. Etiologies were cerebrovascular disease (56%), intracranial bleeding (17%), tumors (14%), infection (9%), and others (6%). Short-term outcome was excellent, with 99.3% in-hospital survival in patients with and 99.4% in patients without neuroimaging, and 97.1% in sICA; 1-year survival in outpatients with neuroimaging was 99.2%, 99.0% in outpatients without, and 88.6% in patients with sICA. Factors associated with sICA were age, emergency severity index (ESI) of 1 or 2, Glasgow coma score (GCS) under 14, focal neurological signs, and a history of malignancy. Prevalence of headache and incidence of sICA were high, but survival after work-up for nontraumatic headache was excellent in the 94% patients without sICA. Due to the incidence of sICA, extensive indication for neuroimaging in headache patients is further warranted, particularly in patients with risk factors.

## 1. Introduction

Headache is a frequent complaint and the majority of the population is occasionally affected [1]. Headache as the reason to present to an emergency department (ED), on the other hand, seems to account for only 1% to 4% of all ED visits [2,3,4,5,6,7]. While causes and outcomes in emergency patients presenting with headache as a “chief complaint” have been reported [8,9,10,11], there is a lack of studies on systematic assessment of headache in an unscreened ED population. We have previously described that headache is among the most prevalent symptoms in ED patients, affecting more than one out of six patients, if patient-reported symptoms are not screened, or filtered by physicians [12]. As the majority of ED patients report more than one symptom [13], and headache patients report a median of four symptoms (often in combination with nonspecific complaints [14,15]), it is highly likely that previous studies have underreported headache, potentially due to inclusion bias and the physician filter. Constructs such as “chief complaint” or “main symptom” heavily rely on several steps of selection and reduction of information [16]—physicians tending to focus on frequent and specific presenting symptoms [17]. This approach may have advantages regarding rapid and highly standardized work-up in clinical settings, but also disadvantages, both in clinical and in research settings. Research on prevalence, outcomes, underlying conditions, and the yield of radiological work-up in headache has rarely been conducted in prospective all-comer populations.

In the last twenty years, the utilization of computed tomography (CT) and magnetic resonance imaging (MRI) for the assessment of headache has more than doubled [18,19], significant intracranial abnormalities decreasing disproportionally [18]. While the main questions on the indications of neuroimaging in traumatic headache have been tackled [20,21,22], the use of neuroimaging in nontraumatic headache is still highly debated. Of all radiological work-up for nontraumatic headache, only about 3% [5,18,23,24] showed significant intracranial abnormalities (sICA). On the higher end, as an example of a possible inclusion bias, a feasibility study including patients in whom “brain CT was required” reported 15% sICA [24]. A well-conducted study, with tight inclusion criteria, reported 6.5% subarachnoid hemorrhage (SAH), endorsing decision rules to lower scan-rates from 83% to between 64% and 74% in typical situations [25]. For various reasons, such as cost and exposure to radiation and contrast agents, positivity rates under 5% may be considered a screening rather than a well-indicated diagnostic procedure. Screening, however, needs to meet higher standards regarding patient safety. Therefore, most authors have demanded more rigorous indications of neuroimaging regarding work-up of headache. Studies investigating risk factors for significant intracranial abnormalities (sICA) have rendered controversial findings. Except for age, no risk factor was identified in a higher number of studies on the use of neuroimaging, and no decision rule has ever been successfully implemented in this field, except for the exclusion of SAH in selected patients [25].

Therefore, we had three aims for the present study: First, to report on the prevalence of nontraumatic headache in a consecutive all-comer ED population. Second, to describe the etiologies responsible for the symptom. Third, to assess the rate of significant intracranial abnormalities (sICA) and to identify potential risk factors of sICA in these patients.

## 2. Methods

### 2.1. Study Design

This is a secondary analysis of a prospective single center all-comer study conducted at the ED of the University Hospital Basel, Switzerland, a tertiary care university hospital with an annual ED census of more than 50,000 patients and a bed capacity of about 700. Data were collected in three periods of consecutive sampling, lasting three to nine weeks. The first was realized between October 21 to November 11 2013, the second one between February 1 and 23 2015, and the last between March 18 and May 5 2019. Three different time points were chosen because of logistic and financial reasons (24/7 study teams) and to minimize seasonal effects by selecting three different seasons. The study protocol was approved by the local ethics committee (www.eknz.ch; identifier 236/13).

### 2.2. Selection of Participants

During the study periods, all patients presenting to the ED were eligible. Pediatric, ophthalmic, and obstetric patients were not included, because they are treated in separate facilities nearby. Patients unwilling or unable to participate, due to intoxication, language problems, severe dementia, unconsciousness, or ongoing life support, were not included. If consent was denied, patients were not included.

### 2.3. Data Collection

For every patient presenting to the ED, an electronic health record (EHR) was opened. For formal triage, an emergency physician or a triage nurse used the German version of the emergency severity index (ESI) [26]. Thereafter, a systematic interview was conducted by the study team. The study team consisted of medical students trained specifically for systematic interviewing. They interviewed every patient at presentation about her/his symptoms, and multiple answers were allowed. They used a questionnaire with 35 symptoms, but no additional questions were asked about duration, onset, and intensity of headache. These properties were identified retrospectively using the chart review methods described below. The data were reviewed by an external institution (Health Care Research Institute, Zürich, Switzerland) digitalized, and anonymized. All patients reporting headache as a symptom were included. Patients were excluded if headache was related to trauma. The definition of trauma-related headache was based on the ICHD3-Classification [27]. In a second step, records of patients who received a head CT and/or MRI were extracted from the radiology database. For all these patients we performed individual chart reviews, based on the electronic health record (EHR: ISMed; ProtecData AH, Boswil, Switzerland). A case report form designed with Microsoft Access 2016 was used to document all results. Two different chart reviewers systematically abstracted all charts of all included patients independently. The predefined protocol contained all criteria defined by experts before chart abstraction. Eleven out of twelve criteria of Worster’s chart review methods [28] were applied. Only blinding to the hypotheses was not possible, as reviewers were aware of the study objectives. The following data were extracted from by chart review: Duration, onset, nausea and/or vomiting, visual disturbance, loss of consciousness, thunderclap type headache, physical activity, Glasgow coma scale (GCS) at presentation, neck stiffness, and focal neurological signs reported by neurologists. In case of disagreement between the two chart reviewers, an expert physician served as a refer.

Radiology reports were categorized by two different neuro-radiologists according to the following three categories: (1) Normal/ consistent finding, (2) insignificant abnormality, (3) significant abnormality. Significant abnormality was defined as a finding requiring therapy, follow-up imaging necessary, or any other clinically important finding, such as cerebrovascular disease. The categorization is shown in Appendix A, Table A1.

Information about all outcomes (hospitalization, intensive care, and mortality) were extracted from the EHR. Hospitalization, according to Swiss law, was defined as at least one overnight stay in a hospital bed. Length of stay (LOS) was defined as the number of days spent in hospital during the index hospitalization. Intensive care was defined as any admission to medical, surgical, or neurosurgical ICUs, or a stroke unit during the index hospitalization. In-hospital mortality was defined as the percentage of patients who died after presenting to the ED without being discharged between admission and death.

### 2.4. Outcomes

The primary outcome of this study is the prevalence of nontraumatic headache in an unselected ED population. Secondary outcomes are the characterization of the population, etiologies, incidence of significant intracranial abnormalities (sICA), and associated risk factors, as well as prognosis, as defined by hospitalization, intensive care, and survival, according to the predefined patient groups.

### 2.5. Statistical Analyses

The statistical analysis was performed using R version 3.6.1 (R-Project, Vienna, Austria). Descriptive statistics are presented as counts and frequencies for categorical data and medians (first quartile, third quartile) for metric variables. Overall *p*-values correspond to the T-test (for means), Kruskall–Wallis test (for median), and chi-squared or exact Fisher test when the expected frequencies is less than 5 in some cell. A *p*-value <0.05 is considered as significant. 

Univariate logistic regression analysis was performed to assess the adjusted odds ratios (OR) with 95% confidence intervals for factors associated with sICA.

## 3. Results

### 3.1. Study Enrolment

During the study period, 14,397 presentations to the ED were registered. Out of 12,562 presentations screened by the study team, a total of 11,269 presentations were included for further analysis. Headache was reported in 1620 presentations, 488 of these presentations were trauma-associated and were therefore excluded. There were 1132 presentations that reported nontraumatic headache. The majority (829 presentations, 73%) did not receive neuroimaging. Of the 1132 nontraumatic headache presentations, 303 (27%) underwent neuroimaging. There were no multiple presentations among these 303 cases. Therefore, 303 patients represented 303 presentations.

A cCT was performed in 198 patients (26 with angiography), a cMRI was performed in 173 patients and in 67 patients, both modalities were used, resulting in 371 examinations (see Figure 1).

Median age in all nontraumatic headache patients was 47 (IQR 33, 63) years. Median age in patients undergoing neuroimaging was 55 (IQR 38, 70) years, and median age for patients without neuroimaging was 44 (IQR 32, 59) years. Patients who underwent neuroimaging were assigned a higher triage score and were older (for details see Table 1).

### 3.2. Significant Intracranial Abnormalities

Of the 303 patients who underwent imaging, 70 (23.1%) had a significant intracranial abnormality (sICA). Fifty-three patients (17.5%) had at least one insignificant abnormality. Normal or consistent findings were described in 174 (57.4%) patients. Six patients (1.9%) had an extracranial abnormality. The majority of the sICA were attributed to the category of cerebrovascular disease (39 patients, 55.7%). In this category, acute and subacute ischemic stroke was the most frequent finding, with 29 of 40 cases (72.5%). High-grade cerebrovascular stenosis without signs of stroke was found in four patients (5.7%), located in the internal carotid artery in three, and in in the posterior cerebral artery in one case. Carotid artery dissection was found in three patients (4.3%), located intracranially in two and extracranially in one case. Aneurysms were found in two patients (2.9%), one in the carotid bifurcation and one in the bifurcation of the arteria cerebri media. Cerebral venous sinus thrombosis was found in two patients (2.9%). Intracranial bleeding was reported in 12 patients (17.1%). Five patients (7.1%) had a subdural hematoma. Intraparenchymal hemorrhage was found in four (5.7%), and subarachnoid hemorrhage in three (4.3%) patients. Tumors were found in 10 patients (14.2%). Three patients (4.3%) had a new malignant tumor (one medulloblastoma, one primary large B-cell lymphoma of the central nervous system, one diffuse astrocytoma grade III). Three (4.3%) had considerable tumor progression (one meningioma, one high-grade glioma, one cerebral metastasis). Two (2.9%) had new cerebral metastases, one (1.4%) a new benign neoplasia (ganglioglioma) and one (1.4%) had a suprasellar nodular lesion of unknown etiology. Central nervous system infections were found in six (8.6%) patients (two meningitis, two encephalitis, one subdural empyema, one shunt infection). The remaining three patients (4.3%) had other findings. One patient had three different significant abnormalities: An acute stroke, an acute left hemispheric subdural hematoma, and an acute occipital intra-parenchymal hemorrhage. As the acute subdural hematoma was the most significant pathology, the other two findings were not listed (see Table 2).

### 3.3. Insignificant Intracranial Abnormalities

The most frequent insignificant abnormalities were acute or chronic sinusitis (26 patients, 49%). Leukoencephalopathy was found in 11 patients, white matter hyperintensities in eight patients, ≤4 microhemorrhages in four patients, a cavernous hemangioma in three patients, and a benign cyst in two patients. Other insignificant abnormalities were found in seven patients (shown in Appendix B, Table A2). Seven patients had more than one insignificant abnormality (53 patients, 60 findings).

### 3.4. Extracranial Abnormality

As cranial CT angiography includes the neck and cranial parts of the chest, extracranial abnormalities were found in six (1.9%) patients: Two patients had micronoduli in the left upper lobe of the lung, one patient had an enlarged para-tracheal lymph node, one had a nodular structure in the anterior-superior mediastinum, another one had a hypo-dense lesion in the thyroid gland, and one suffered from struma nodosa.

### 3.5. Factors Associated with Intracranial Abnormality

Univariate analyses showed significant correlations between age, high acuity ESI scores (ESI 1 and ESI 2), Glasgow coma scale (GCS) <14, focal neurological signs, malignant tumors, and significant intracerebral abnormalities (sICA). Dizziness was the only factor correlating negatively with a sICA. 

No association was found with sICA and the following features: Gender, blood pressure, temperature, feeling feverish, cough, weakness, fatigue, thunderclap headache, nausea or vomiting, visual disturbance, loss of consciousness, physical activity at onset of headache, duration of headache, onset, neck stiffness, hypertension, diabetes mellitus, obesity, or immunosuppression (for details see Table 3).

### 3.6. Outcomes

Of all patients with nontraumatic headache undergoing neuroimaging, 176 (58.1%) patients were hospitalized and 127 (41.9%) received outpatient follow-up. Median LOS was four (IQR 2, 10) days. Intensive care was deemed necessary in 30 (9.9%) patients. Two patients (0.7%) died during the index hospital stay. Both of them had a sICA; therefore, 2.9% of patients with sICA died during the index hospital stay.

In the group with nontraumatic headache without imaging, 201 (24%) patients were hospitalized. Median LOS was five (IQR 2, 8) days, 14 patients (1.7%) were transferred to intensive care, and five patients died during index hospitalization (0.6%).

The groups (with and without imaging) differed significantly (*p* < 0.001) regarding hospitalization and intensive care (see Table 4).

One-year survival in outpatients was as follows: In the group with neuroimaging, all but one patient survived (99.2%), in the group without imaging, all but five patients survived (99.1%). In patients with sICA, all but eight patients survived (88.6%) (see Table 5).

## 4. Discussion

The main findings of this study are the high prevalence of nontraumatic headache in an unselected emergency cohort, the high rate of significant intracranial abnormality, and high acuity triage categories as well as a history of malignancy as risk factors for such findings. Other risk factors, such as age, focal neurological signs, and abnormal GCS, have previously been described [2,3,4,6,18,23,29,30,31,32]. The only negatively correlating factor was dizziness, a typical nonspecific complaint, commonly associated with benign outcome. It was shown that, e.g., migraine and primary headache are often associated with dizziness [33]. On the other hand, patients with dizziness very rarely suffered from brain tumors or other serious brain pathologies [34]. These findings are corroborated with our data. The excellent survival in patients with headache was already shown in Swedish cohorts [35,36]. In detail, the prevalence of headache and the prevalence of sICA is considerably higher as compared to other studies [2,5,37], possible reasons being the higher median age and the lack of selection for certain features in our cohort [38,39,40]. Other studies did not examine emergency patients [37,41,42], which makes comparisons very difficult. Most of the differences may be explained by the inclusion procedure, as all our patients were systematically interviewed at presentation. Therefore, there was no filtering of information by physicians. Although this initial filtering has been successfully applied for centuries [16], its general use may hamper symptom-oriented research, because a potentially important part of the information cannot be analyzed. As headache is typically a symptom among others [14], it has a higher probability to go unrecorded than, e.g., leg pain, the latter often reported as an isolated symptom. The threefold expansion of the prevalence of headache by systematic interviewing may have resulted in over-reporting, watering down the prevalence of important findings. However, this would have resulted in lower sICA than reported in more highly selected populations [8,43]. Interestingly, this was not the case, as we have identified more sICA than other prospective ED cohorts [8,30]. More than half of our headache patients did not undergo neuroimaging, our scan rate being comparable to other cohorts. Due to the lack of decision rules (except for SAH) on the need of imaging in acute headache [44,45,46], study inclusion criteria vary widely. Among over 5000 reports, only four have been highlighted by the recent ACEP policy [47]—three concerning SAH exclusion rules in selected patients [25,45] and our prospective study on the use of copeptin [48]. Due to the lack of prospective validation and the exclusion of only 91% sICA using the lowest cut-off value, a change of clinical practice has not occurred. Carpenter’s meta-analysis showed that none of 17 clinical variables had test characteristics powerful enough to reliably rule SAH in or out. SAH accounted for less than 5% of all sICA in our population and in comparable studies, but it has received disproportionate attention in the literature. On the other hand, the high overall incidence of sICA in our patients showed that the selection of patients to undergo neuroimaging was adequate. In a comparably difficult situation, pulmonary embolism (PE) rule-out, the incidence of significant findings in CT pulmonary angiography in our ED was also over 15%, indicating a generally conservative indication of CT [49] in our ED. The comparison between such diverse entities is interesting from the point of view of the hit-to-miss ratio in potentially lethal situations. In the case of PE, some authors have claimed that the hit-to-miss ratio should exceed 10% in order not to drop down to screening range, where inordinate economic and safety considerations may come into place. In our cohort, the scan rate of 27% with a hit (positivity) rate of 23% seems adequate.

It is not surprising that risk factors, such as age, focal neurological signs, abnormal GCS, and previously diagnosed malignant tumors were identified as being associated with sICA. First, most were previously described, and second, all of these features may be overrepresented in a population “needed to be scanned”. In fact, patients with neurological signs, though often subtle and detected by neurologists, and abnormal GCS were included, which was not the case in other studies [38,39,40,50,51]. While the influence of age on sICA seems obvious, the newly described association of high acuity ESI with sICA needs to be highlighted. Particularly ESI 2, representing a large part of emergency patients [52] seems to be prognostic for sICA. This finding may have an influence on clinical practice regarding indication for neuroimaging in nontraumatic headache. Several consequences should be considered based on the presented data: First, headache seems to be underreported in other studies, as the incidence was higher in our cohort as compared to other cohorts. Therefore, a more systematic assessment of headache seems to be needed. Second, the diagnostic yield of imaging was higher than in other studies, suggesting a liberal indication of imaging. Third, differential diagnosis was extremely broad, and subacute stroke should be considered in unclear headache in older patients.

### Limitations

There are several limitations to this study. First, this is a single center study, and both the population, and the practice guidelines are local, which certainly limits external validity. Second, it is a secondary analysis of prospectively collected data. However, we have collected the data according to our long-term venture, symptom-oriented research [12,16]—the advantage being the assessment of an all-comer cohort with a minimal inclusion bias. Third, although all symptoms of all patients were prospectively and systematically documented by a study team around the clock, headache was not interrogated in depth, e.g., we did not predefine factors, such as vision disturbance or descriptions of thunderclap, and we left it up to patients to mention such symptoms in order to minimize suggestive interviewing or overreporting. Such descriptors were taken down as “other symptoms”, but we cannot exclude that some were not recorded, as they were not systematically assessed. In order to close this gap, each case underwent a detailed and standardized chart review. Another limitation is the fact that we cannot guarantee that every work-up was conducted to our own protocols (medstandards.com), based on the neuroimaging guidelines of the American Headache Society and American Academy of Neurology. However, since our protocols are extensively used, we believe that adherence to these standards is high and that the monocentric approach is an advantage in this regard. Further, the inclusion of all patients, including patients with neurological signs and symptoms, may have included underlying etiologies that were not described in other prospective studies. However, such signs may be subtle and are often missed in older patients, particularly if they present with generalized weakness [15]. Even if we would have excluded all cerebrovascular etiologies, differential diagnosis of nontraumatic headache would have remained very broad, and, e.g., SAH would have represented only 10% of all sICA. Finally, the outcome may be worse than reported, as the majority of patients was discharged. However, both our non-scanned population as well as our out-patient population had an excellent 1-year survival with acceptable follow-up rates (see Table 5). Most importantly, this is an exploratory study aiming to test the hypothesis that systematic history taking is feasible and that certain subgroups of ED patients could be monitored focusing on outcomes and the use of imaging with the associated hit-to-miss ratios. The study was a mere observation and was not powered to obtain risk categories or even a rule-out algorithm. However, the approach shown may be used in the future in order to risk-stratify by observing a large set of patients at multiple sites using systematic history taking.

## 5. Conclusions

Taken together, this is an observation of patients with nontraumatic headache showing a rather high prevalence of sICA and a broad distribution of etiologies in patients undergoing neuroimaging. While the risk factors identified (age, high acuity triage, focal neurological signs, pathologic GCS, and a history of malignant tumors) may increase the probability of sICA in our population, the absence of these risk factors does not rule out sICA to a clinically useful degree. Therefore, extensive indications for neuroimaging in headache patients is warranted, particularly in patients with risk factors. However, hit-to-miss ratios should be monitored, as rates <10% may indicate a necessity for quality control, and possibly a more conservative indication for neuroimaging in nontraumatic headache.

## Figures and Tables

**Figure 1 jcm-09-02621-f001:**
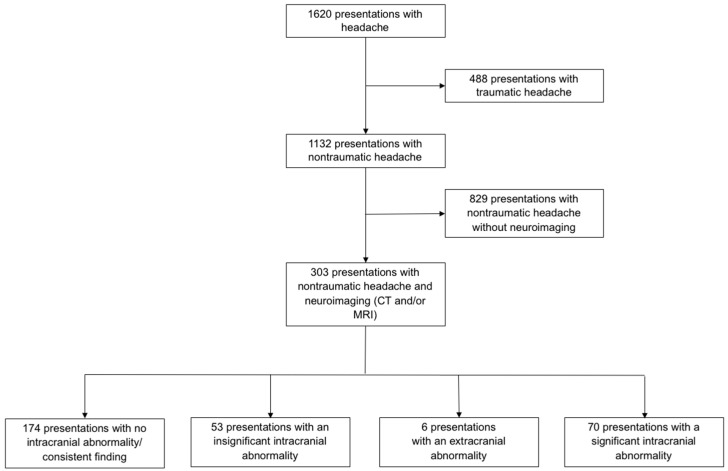
Study population.

**Table 1 jcm-09-02621-t001:** Baseline characteristics.

	All Nontraumatic Headache	Nontraumatic Headache without Imaging	Nontraumatic Headache with Imaging
Presentations, *n* (%)	1132 (100)	829 (73.3)	303 (26.7)
Age (years), median (Q1, Q3)	47 (33, 63) *	44 (32, 59)	55 (38, 70)
Gender (female), *n* (%)	646 (57.1)	469 (56.6)	178 (58.8)
ESI, n (%)	*		
1	9 (0.8)	4 (0.5)	5 (1.7)
2	342 (30.2)	210 (25.3)	132 (43.6)
3	533 (47.1)	381 (46.0)	152 (50.2)
4	239 (21.1)	225 (27.1)	14 (4.6)
5	9 (0.8)	9 (1.1)	0 (0)

* *p* < 0.001, *p*-value refers to comparison between groups “nontraumatic headache without imaging” and “nontraumatic headache with imaging”. ESI (Emergency Severity Index) category is the urgency level assigned at triage.

**Table 2 jcm-09-02621-t002:** Patients with significant intracranial abnormalities on computed tomography (CT)/magnetic resonance imaging (MRI)**.**

Diagnosis	Patients
Cerebrovascular disease (*n* = 39)	
- Acute or subacute stroke	28
- High grade arterial stenosis (three in internal carotid artery, one in posterior cerebral artery)	4
- Carotid artery dissection (two in internal carotid artery, one in external carotid artery)	3
- Aneurysm (one in the bifurcation of carotid artery, one in the bifurcation of middle cerebral artery)	2
- Sinus venous thrombosis	2
Intracranial bleed (*n* = 12)	
- Acute or chronic or acute on chronic subdural hematoma	5
- Intraparenchymal hemorrhage	4
- Subarachnoidal hemorrhage	3
Malignancy (*n* = 10)	
- Malignant neoplasia (one medullablastoma, one primary large B-cell lymphoma of the central nervous system, one diffuse astrocytoma grade III)	3
- Tumor progression (one meningioma, one high-grade glioma, one cerebral metastasis)	3
- Cerebral metastasis	2
- Benign neoplasia (ganglioglioma)	1
- Supra-sellar nodular lesion of unknown etiology	1
Central nervous system (CNS) infection (*n* = 6)	
- Viral encephalitis	2
- Meningitis	2
- Subdural empyema	1
- Shunt infection	1
Others (*n* = 3)	
- Pseudotumor cerebri	1
- Soft-tissue phlegmon	1
- Space-occupying hygroma	1
Total	70

**Table 3 jcm-09-02621-t003:** Univariate analysis of features predictive of significant intracranial abnormality.

		All Patients with Imaging	No New Finding/ Insignificant Abnormality	Significant Abnormality	OR [95% CI]
		303	233	70	
	median (Q1, Q3)				
Systolic blood pressure, (mmHg)		135 (122;154)	133 (121;153)	145 (125;159)	1.01 (1.00;1.02)
Diastolic blood pressure, (mmHg)		81 (73;90)	81 (73;90)	80 (71;91)	0.99 (0.98;1.01)
Body temperature, (C°)		36.8 (36.4;37.1)	36.8 (36.4;37.0)	36.7 (36.2;37.3)	0.95 (0.63;1.43)
Age, (years)		55 (38;70)	51 (36;66)	64.5 (52;79)	1.03 (1.02;1.05) ***
	*n* (%)				
Gender	Male	125 (41.3)	95 (40.8)	30 (42.9)	Ref.
	Female	178 (58.7)	138 (59.2)	40 (57.1)	0.92 (0.53;1.59)
ESI	1	5 (1.7)	2 (0.9)	3 (4.3)	7.67 (1.11;68.8) *
	2	132 (43.6)	91 (39.1)	41 (58.6)	2.39 (1.36;4.29) **
	3	152 (50)	128 (54.9)	24 (34.3)	Ref.
	4	14 (4.6)	12 (5.2)	2 (2.9)	0.94 (0.13;3.82)
Feeling feverish	No	290 (95.7)	223 (95.7)	67 (95.7)	Ref.
	Yes	13 (4.3)	10 (4.3)	3 (4.3)	1.03 (0.22;3.56)
Cough	No	288 (95.0)	219 (94.0)	69 (98.6)	Ref.
	Yes	15 (5.0)	14 (6.0)	1 (1.4)	0.26 (0.01;1.32)
Weakness	No	244 (80.5)	192 (82.4)	52 (74.3)	Ref.
	Yes	59 (19.5)	41 (17.6)	18 (25.7)	1.62 (0.84;3;04)
Fatigue	No	266 (87.8)	208 (89.3)	52 (74.3)	Ref.
	Yes	37 (12.2)	25 (10.7)	12 (17.1)	1.73 (0.79;3.60)
Dizziness	No	185 (61.1)	135 (57.9)	50 (71.4)	Ref.
	Yes	118 (38.9)	98 (42.1)	20 (28.6)	0.55 (0.30;0.98) *
Thunderclap headache	No	282 (93.1)	216 (92.7)	66 (94.3)	Ref.
	Yes	21 (6.93)	17 (7.3)	4 (5.7)	0.79 (0.22;2.25)
Duration	≥24 h	160 (52.8)	119 (51.1)	41 (58.6)	Ref.
	<24 h	143 (47.2)	114 (48.9)	29 (41.4)	0.74 (0.43;1.27)
Nausea and/or vomiting	No	194 (64)	152 (25.2)	42 (60.0)	Ref.
	Yes	109 (36)	81 (34.8)	28 (40.0)	1.25 (0.72;2.16)
Visual disturbance	No	246 (81.2)	185 (79.4)	61 (87.1)	Ref.
	Yes	57 (18.8)	48 (20.6)	9 (12.9)	0.58 (0.25;1.20)
Loss of consciousness	No	277 (91.4)	214 (91.8)	63 (90.0)	Ref.
	Yes	26 (8.6)	19 (8.2)	7 (10.0)	1.27 (0.47;3.05)
Onset	Subtle	259 (85.5)	195 (83.7)	64 (91.4)	Ref.
	Sudden	44 (14.5)	38 (16.3)	6 (8.6)	1.27 (0.47;3.05)
Physical activity	No	298 (98.3)	228 (97.9)	70 (100)	Ref.
	Yes	5 (1.7)	5 (2.1)	0 (0.0)	not estimated
Glasgow coma scale	15	273 (90.1)	217 (93.1)	56 (80)	Ref.
	14	18 (5.9)	12 (5.2)	6 (8.6)	1.96 (0.64;5.34)
	< 14	12 (3.9)	4 (1.7)	8 (11.4)	7.54 (2.24;30.1) **
Focal neurological signs	No	224 (73.9)	191 (82.0)	33 (47.1)	Ref.
	Yes	79 (26.1)	42 (18.0)	37 (52.9)	5.06 (2.85;9.08) ***
Neck stiffness	No	295 (97.4)	227 (97.4)	68 (97.1)	Ref.
	Yes	8 (2.6)	6 (2.58)	2 (2.9)	1.17 (0.15;5.39)
Hypertension	No	205 (67.7)	164 (70.4)	41 (58.6)	Ref.
	Yes	98 (32.3)	69 (29.6)	29 (41.4)	1.68 (0.96;2.92)
Diabetes mellitus	No	268 (88.4)	210 (90.1)	58 (82.9)	Ref.
	Yes	35 (11.6)	23 (9.9)	12 (17.1)	1.89 (0.86;4.00)
Obesity	No	278 (91.7)	214 (91.8)	64 (91.4)	Ref.
	Yes	25 (8.3)	19 (8.2)	6 (8.6)	1.07 (0.37;2.68)
Malignancy	No	270 (89.1)	218 (93.6)	52 (74.3)	Ref.
	Yes	33 (10.9)	15 (6.4)	18 (25.7)	4.99 (2.35;10.7) ***
Immunosuppression	No	293 (96.7)	226 (97.0)	67 (95.7)	Ref.
	Yes	10 (3.3)	7 (3.0)	3 (4.3)	1.48 (0.30;5.65)

OR = Odds ratio; CI = Confidence Interval; * *p* < 0.05, ** *p* < 0.01, *** *p* < 0.001.

**Table 4 jcm-09-02621-t004:** Outcomes.

	All Nontraumatic Headache (*n* = 1132)	Nontraumatic Headache without Imaging (*n* = 829)	Nontraumatic Headache with Imaging (*n* = 303)
Hospitalization, *n* (%)	377 (33.3) *	201 (24.3)	176 (58.1)
LOS, median (Q1, Q3)	5 (2, 9)	5 (2, 8)	4 (2, 10)
ICU, *n* (%)	44 (3.9) *	14 (1.69)	30 (9.9)
Mortality (in-hospital), *n* (%)	7 (0.6)	5 (0.6)	2 (0.7)

* *p* < 0.001, *p*-value refers to the comparison between “nontraumatic headache without imaging” and “nontraumatic headache with imaging” groups.

**Table 5 jcm-09-02621-t005:** One-year follow-up.

	Outpatients with Imaging n = 127	Outpatients without Imaging *n* = 555	Patients with sICA *n* = 70
One-Year Survival, n(%)			
yes	124 (99.2)	510 (99.1)	62 (88.6)
no	1 (0.8)	5 (0.9)	8 (11.4)
Lost to Follow-up, *n* (%)	2 (1.6)	40 (7.2)	0 (0)

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
