# Peer review of "Nontraumatic Headache in Adult Emergency Patients: Prevalence, Etiologies, and Radiological Findings"

_jcm, 2020, doi:10.3390/jcm9082621_

Round 1

Reviewer 1 Report

We do need more neuroimaging studies and descriptions of presenting symptoms to the Emergency Room and am very impressed that this team has worked on this.

Would recommend in title or background of abstract that this focuses on adults instead of just in the methods for readers, as pediatric headaches are a different ballgame when it comes to presenting symptoms to the emergency room.

Interesting to not find any associations with certain symptoms to significant vs. nonsignificant intracranial abnormalities; it would be great for future studies to study these symptoms with each condition diagnosed via neuroimaging to help ED providers narrow down presenting symptoms.

Methods: May have been relevant to include ICHD3 diagnoses as part of methods to assess percentage of individuals with significant abnormality diagnosed with migraine vs. TTH vs. cluster headache for example. This would add to the headache field as neuroimaging guidelines are definitely not perfect and can miss these diagnoses presented in this paper.

In the discussion, would highly recommend adding reference to the American Headache Society and American Academy of Neurology neuroimaging guidelines and quality of care:  https://www.aan.com/siteassets/home-page/policy-and-guidelines/quality/quality-measures/14headachemeasureset_pg.pdf

Just a few formatting changes  needed as new paragraphs are tabbed instead of a new paragraph, but may have been formatted unusually when converting to pdf.

Reviewer 2 Report

The authors conducted a population-based study that comes to the emergency room selected for the non-traumatic 'headache' symptom
They observed that the prevalence of this problem is about 10% of the patients presenting in the ER, the most frequent cause is subacute stroke.

The paper is generally well oriented and straightforward for me. The first point that should be further discussed is the role of headache: how can we manage to be more accurate in its evaluation after this paper? Can helpers somehow select a cluster of conditions that seen together allow you to orientate the imaging correctly?
The second important aspect is related to the role of imaging: how much could we identify using the rmn in ER instead of CT scan?

Minor points

Abstract: to be summarized

Introduction
Lines 79-86: it is probably a typo during the construction of the work. However, we invite the authors to be very careful before submitting their work in order to facilitate the referee's work

Methods: The choice of three periods of time spaced from a few years is a fairly important limitation of the work: how are the authors sure that in the meantime attitudes, protocol or internal indications have not changed

Results: dizziness correlates negatively; this finding deserves a more detailed comment

Round 2

Reviewer 2 Report

No further comments